# Investigation of Antifungal Properties of Synthetic Dimethyl-4-Bromo-1-(Substituted Benzoyl) Pyrrolo[1,2-a] Quinoline-2,3-Dicarboxylates Analogues: Molecular Docking Studies and Conceptual DFT-Based Chemical Reactivity Descriptors and Pharmacokinetics Evaluation

**DOI:** 10.3390/molecules26092722

**Published:** 2021-05-06

**Authors:** Vijayakumar Uppar, Sandeep Chandrashekharappa, Chandan Shivamallu, Sushma P, Shiva Prasad Kollur, Joaquín Ortega-Castro, Juan Frau, Norma Flores-Holguín, Atiyaparveen I. Basarikatti, Mallikarjun Chougala, Mrudula Mohan M, Govindappa Banuprakash, Katharigatta N. Venugopala, Belakatte P. Nandeshwarappa, Ravindra Veerapur, Abdulaziz A. Al-Kheraif, Abdallah M. Elgorban, Asad Syed, Kiran K. Mudnakudu-Nagaraju, Basavaraj Padmashali, Daniel Glossman-Mitnik

**Affiliations:** 1Department of Chemistry, School of Basic Science, Rani Channamma University, Belagavi 591156, Karnataka, India; vijay.uppar@gmail.com (V.U.); atiyabasarikatti59@gmail.com (A.I.B.); 2Institute for Stem Cell Science and Regenerative Medicine, NCBS, TIFR, GKVK-Campus Bellary road, Bengaluru 560065, Karnataka, India; sandeep_m7@rediffmail.com; 3Department of Medicinal Chemistry, National Institute of Pharmaceutical Education and Research (NIPER) Raebareli, Lucknow (UP) 226002, India; sushmap@jssuni.edu.in; 4Department of Biotechnology & Bioinformatics, Faculty of Life Sciences, JSS Academy of Higher Education and Research, Mysore 570015, Karnataka, India; chandans@jssuni.edu.in (C.S.); mrudula51219@gmail.com (M.M.M.); 5Department of Sciences, Amrita School of Arts and Sciences, Amrita Vishwa Vidyapeetham, Mysuru 570026, Karnataka, India; shivachemist@gmail.com; 6Departament de Química, Universitat de les Illes Balears, E-07122 Palma de Malllorca, Spain; joaquin.castro@uib.es (J.O.-C.); juan.frau@uib.es (J.F.); 7Laboratorio Virtual NANOCOSMOS, Departamento de Medio Ambiente y Energía, Centro de Investigación en Materiales Avanzados, Chihuahua, Chih 31136, Mexico; norma.flores@cimav.edu.mx (N.F.-H.); dglossman@gmail.com (D.G.-M.); 8Department of Biotechnology, JSS College of Arts, Commerce and Science (Autonomous), Mysore 570025, Karnataka, India; mallikarjun100@gmail.com; 9Department of Chemistry, SJB Institute of Technology, Bengaluru 560060, Kengeri, India; gbanuprakash@sjbit.edu.in (G.B.); jayadev@sjbit.edu.in (J.); 10Department of Pharmaceutical Sciences, College of Clinical Pharmacy, King Faisal University, Al-Ahsa 31982, Saudi Arabia; kvenugopala@kfu.edu.sa; 11Department of Studies in Chemistry, Shivagangothri, Davangere University, Davangere 577007, Karnataka, India; belakatte@davangereuniversity.ac.in; 12Department of Metallurgy and Materials Engineering, Malawi Institute of Technology, Malawi University of Science and Technology, P.O. Box-5916 Limbe, Malawi; rveerapur@must.ac.mw; 13Dental Biomaterials Research Chair, Dental Health Department, College of Applied Medical Sciences, King Saud University, P.O. Box 10219, Riyadh 11433, Saudi Arabia; aalkhuraif@ksu.edu.sa; 14Department of Botany and Microbiology, College of Science, King Saud University, P.O. Box 2455, Riyadh 11451, Saudi Arabia; aelgorban@ksu.edu.sa (A.M.E.); assyed@ksu.edu.sa (A.S.); 15Department of Biotechnology and Food Technology, Faculty of Applied Sciences, Durban University of Technology, Durban 4001, South Africa

**Keywords:** BQ molecules, pyrrolo[1,2-a] quinolone, indolizines, molecular docking, DFT—density functional theory, conceptual DFT, *Candida albicans*, antifungal drug

## Abstract

*Candida albicans*, an opportunistic fungal pathogen, frequently colonizes immune-compromised patients and causes mild to severe systemic reactions. Only few antifungal drugs are currently in use for therapeutic treatment. However, evolution of a drug-resistant *C. albicans* fungal pathogen is of major concern in the treatment of patients, hence the clinical need for novel drug design and development. In this study, in vitro *screening* of novel putative pyrrolo[1,2-a]quinoline derivatives as the lead drug targets and in silico prediction of the binding potential of these lead molecules against *C. albicans* pathogenic proteins, such as secreted aspartic protease 3 (SAP3; 2H6T), surface protein β-glucanase (3N9K) and sterol 14-alpha demethylase (5TZ1), were carried out by molecular docking analyses. Further, biological activity-based QSAR and theoretical pharmacokinetic analysis were analyzed. Here, in vitro screening of novel analogue derivatives as drug targets against *C. albicans* showed inhibitory potential in the concentration of 0.4 µg for BQ-06, 07 and 08, 0.8 µg for BQ-01, 03, and 05, 1.6 µg for BQ-04 and 12.5 µg for BQ-02 in comparison to the standard antifungal drug fluconazole in the concentration of 30 µg. Further, in silico analysis of BQ-01, 03, 05 and 07 analogues docked on chimeric 2H6T, 3N9K and 5TZ1 revealed that these analogues show potential binding affinity, which is different from the therapeutic antifungal drug fluconazole. In addition, these molecules possess good drug-like properties based on the determination of conceptual Density Functional Theory (DFT)-based descriptors, QSAR and pharmacokinetics. Thus, the study offers significant insight into employing pyrrolo[1,2-a]quinoline analogues as novel antifungal agents against *C. albicans* that warrants further investigation.

## 1. Introduction

Globally, *Candida albicans (C. albicans)* is a genus of yeast that frequently causes nosocomial fungal infection in patients with a defined clinical condition—i.e., invasive candidiasis and candidemia [1,2]. Being an opportunistic fungal pathogen, *C. albicans* asymptomatically colonized mucosal surfaces of the skin, vagina, mouth, and intestine in patients suffering from invasive candidiasis as well as systemically colonizes the bloodstream in candidemia patients and poses a major therapeutic challenge in medically and immune-compromised patients, with a mortality rate of 30–40% [3,4,5,6]. Further, *C. albicans* frequently infects humans with pre-existing conditions due to overuse of broad-spectrum antibiotics and immunosuppressants, surgical intervention, diabetes, prolonged hospitalization, and radio- as well as chemotherapy (in case of malignancy) [7,8]. Until now, reports suggested that only few classes of potential antifungal drugs such as azoles, echinocandins, and polyenes are on demand for the treatment of fungal infections and that multidrug resistance fungal pathogens are increasing worldwide along with adverse as well as cytotoxic effects [9,10,11].

The innate immune response predominates in regulating *C. albicans* fungal infectivity in normal individuals [12]. However, in the state of immune dysfunction, fungus invasion results due to switching of yeast to hyphal growth formations, which enter the human body via pathogen recognition factors present on various immune cells, which induces immune reactivity causing a wide range of symptoms from superficial to major life-threatening systemic reactions [3]. However, the number of antifungal drugs in treatment are very limited [13] and therefore, new therapies for fungal infection are of public concern as drug-resistant *C. albicans* are increasing worldwide [9,14,15].

However, in organic chemistry, the maximum heterocyclic compounds possess significant biological activity [16,17,18,19,20,21,22,23,24,25,26,27,28,29,30,31,32,33,34,35,36]. Recent reports suggest that indolizine is a very attractive molecule among the heterocyclic compounds and possesses two fused ring structures with carbon and nitrogen bridging atoms. Further, the numerous indolizine derivatives show potential biological activities [16,21,22,23,24,25,26,27,28,29,30,31,32,33,34,35,36,37,38,39,40,41], which includes antioxidant [42,43,44], antimicrobial [42,43,44,45,46], anti-inflammatory [47], anticancer [48,49] and antituberculosis [50] activities.

Pyrrolo[1,2-a]quinoline and pyrrolo[1,2-a]isoquinoline are family members of indolizine that are also known as 5,6-benzo-fused indolizine and 7,8-benzo-fused indolizine, respectively [51]. The series of these analogues has shown remarkable biological activity so far [52]. For instance, recent publications from V. Uppar et al. (2020) and others shows that pyrrolo[1,2-a]quinoline derivatives effectively inhibit the activity of microbial pathogens such as bacteria (both Gram-positive and Gram-negative) [44,53,54,55,56], and fungus [57], followed by various disease conditions such as cancer [58], malaria [59], inflammation [60] and Alzheimer disease [61]. In addition, these derivatives possess antioxidant activity [44,54,55,56]. Further, ethyl-1-(subbenzoyl)-5-methylpyrrolo[1,2-a]quinoline-3-carboxylate and dimethyl 1-(subbenzoyl)-5-methylpyrrolo[1,2-a]quinoline-2,3-dicarboxylate derivatives have been tested for their larvicidal activity against Anopheles arabiensis [56]. Collectively, the broad-spectrum nature of the compounds intensified further to investigate whether newly synthesized derivatives target *C. albicans* fungal pathogens. Herein, the synthesized dimethyl-4-bromo-1-(4-substituted benzoyl) pyrrolo[1,2-a]quinoline-2,3-dicarboxylate methyl 4-bromo-1-(4-sub-benzoyl)pyrrolo[1,2-a]quinoline-3-carboxylate 1a-n (Scheme 1) [55], a novel analogue derivative (Figure 1), was tested for its inhibitory activity against C. *albicans* in vitro. As shown in Figure 1, these derivatives consist of methyl ester groups present at the second and third positions (highlighted) of BQ-01, 03, 05 and 07, and only one ester group at the third position of BQ-02, 04, 06 and 08.

Furthermore, chemoinformatic methods are utilized for the study and organization of chemical information, a tool widely used for the development of new medical drugs in the pharmaceutical industry. Here, chemoinformatics are applied for the prediction of the molecular properties of many systems based on the knowledge of previously studied molecular structures and computational displaying of the same by considering the close relationship between biological data and organic structures and energies. The literature suggests that the molecular descriptors can be related to molecular properties [62].

In the present study, compounds tested in vitro were anaylzed to elucidate their mechanisms of action by employing in silico molecular docking calculations. In addition, the present work reports the information on the chemical reactivity of the molecules under study through Density Functional Theory (DFT) derived concepts, which shows correlation of their biological activity [63,64,65,66,67,68,69]. Further, the chemical reactivity properties of these molecules were further analyzed using global and local descriptors by extracting the information about their biological reactivity for designing new medical drugs [70,71,72,73,74].

## 2. Results and Discussion

### 2.1. Analysis of MIC against C. albicans

The inhibitory potential of pyrrolo [1, 2-a]quinoline derivatives targeted against *C. albicans* were observed after 24 h of incubation. The derivatives BQ-06, 07 and 08 show the highest minimum inhibitory concentrations (MICs) at 0.4 µg/mL, whereas BQ-01, 03, and 05 have MICs of 0.8 µg/mL. Further, the BQ-04 derivative possesses an MIC of 1.6 µg/mL, followed by BQ-12 which causes inhibition of growth of *C. albicans* at 12.5 µg/mL (Table 1).

### 2.2. Molecular Docking

Binding orientation of pyrrolo[1,2-a] quinoline on target proteins 2H6T, 3N9K and 5TZ1 from C *albicans* were assessed by molecular docking analysis. Out of the eight pyrrolo[1, 2-a] quinoline derivatives, BQ-01, 03, 05, and 07 were used for molecular docking analysis. These molecules possess common methyl ester groups at the second and third positions. However, BQ-02, 04, 06 and 08 consist of a common hydrogen group at the second position and ethyl ester at third position. The results obtained for docking between the macromolecule and ligand show eight poses of the ligand molecule that is docked at the macromolecule (protein). The confirmation of pose consisting of the least binding affinity was saved in .pdb format and their interactions were studied using Pymol software (The PyMOL Molecular Graphics System, Version 2.0 Scrödinger, LLC). The amino acid pocket residues of the macromolecule interaction with the ligand were analyzed for the bonded (H-bonds) and nonbonded interactions (hydrophobic interactions) and were later visualized by using Pymol software. The formation of the number of hydrogen bonds (H-bonds) forming amino acid residues and binding free energy of the interaction between protein targets and ligands obtained are shown in Table 2. Further, the number of hydrogen bonds formed were identified for the ligand molecules BQ-03, BQ-05, and BQ-07 (6, 5, and 6 hydrogen bonds, respectively) against C. *albicans* protein 2H6T, which shows a stronger binding interaction than the drug candidate (four hydrogen bonds). The H-bond forming amino acid residues between the ligand molecules BQ-01, 03, 05, and 07 and the protein target 2H6T were identified as THR-221 and THR-222. Further, BQ-03 and 05 also actively interact with THR-221 and THR-222 along with GLY-220, whereas BQ-07 interacts with THR-221 and THR-222 along with VAL-12. Moreover, these molecules exhibit potential binding affinity to the amino acid residues, which possess different interaction sites compared to that of the therapeutic antifungal drug fluconazole. However, in the cases of C. *albicans* proteins 3N9K and 5TZ1, the number of hydrogen bonds formed with the ligands were less than or equal to fluconazole. Interestingly, the H-bond forming amino acid residues identified by the interaction analyses were different from that of fluconazole, which indicates novel binding interaction of the ligand molecules to the macromolecular target. These results offers pyrrolo[1,2-a]quinoline analogues as putative novel targets against drug-resistant C. *albicans* fungal pathogens. 

Further, the ligands and macromolecular targets (3N9K and 5TZ1) possess stronger binding free energy than fluconazole, as shown in Table 2. Among the tested ligands, BQ-03 shows binding free energies of −7.4, −9.7 and −9.1 kcal/mol against 2H6T, 3N9K and 5TZ1, respectively. The binding interaction between target proteins 3N9K and 5TZ1 and all the tested ligands (BQ-01, BQ-03, BQ-05, and BQ-07) showed ∆G values that are higher than fluconazole and it reveals that these compounds possess a stronger affinity towards the protein targets 3N9K and 5TZ1 than the drug, indicating that the interaction between the ligand molecule and the protein is possible and thermodynamically favorable. Further, a high number of hydrogen bonds were formed with ligand-2H6T interactions; however, strong binding affinity in terms of binding free energy was observed with ligand-3N9K, and ligand-5TZ1 interactions. These results show the presence of nonbonded interactions along with hydrogen bonds in ligand-3N9K and ligand-5TZ1 interactions. The strong binding interaction of ligands BQ-01, 03, 05, and 07, compared to fluconazole, to protein targets 3N9K and 5TZ1, are a good platform for further development of a novel drug target against *C albicans*. The free energy change upon binding of protein–ligand interaction has been shown as follows:

Fluconazole interacted with 2H6T with a binding affinity of −7.5 kcal/mol, while it showed −7.8 and −7.2 kcal/mol binding energy when interacting with 3N9K and 5TZ1, respectively.

The BQ-01 ligand interacted with 2H6T with a binding affinity of −6.5 (kcal/mol), while it showed −9.4 (kcal/mol) and −8.9 (kcal/mol) binding energies when interacting with 3N9K and 5TZ1, respectively.

The BQ-03 ligand interacted with 2H6T with a binding affinity of −7.4 (kcal/mol), while it showed −9.7 (kcal/mol) and −9.1 (kcal/mol) binding energies when interacting with 3N9K and 5TZ1, respectively.

The BQ-05 ligand interacted with 2H6T with a binding affinity of −7.2 (kcal/mol), while it showed −9.4 (kcal/mol) and −8.6 (kcal/mol) binding energies when interacting with 3N9K and 5TZ1, respectively.

The BQ-07 ligand interacted with 2H6T with a binding affinity of −6.8 (kcal/mol), while it showed −9.2 (kcal/mol) and −8.7 (kcal/mol) binding energies when interacting with 3N9K and 5TZ1, respectively. 

Further, the docking result confirms that the drug fluconazole (Figure 2) or ligand molecules BQ-01 (Figure 3), 03 (Figure 4), 05 (Figure 5) and 07 (Figure 6) were held in the active pocket of the C. *albicans* fungal proteins 2H6T, 3N9K or 5TZ1, respectively.

### 2.3. Determination of the Conceptual DFT Reactivity Descriptors of the Molecules and their Related Pharmacokinetics

The determination of conformers of the molecules considered in the current study was performed by utilizing various software described elsewhere, as described in Section 3.5 [75,76,77,78,79,80,81,82,83,84,85,86,87,88,89,90,91]. Further, the results obtained from the model chemistry consider the MN12SX screened-exchange density function [81] together with the Def2TZVP basis set [82,83], and in all cases the charge of the molecules was equal to zero while the radical anion and cation were considered in the doublet spin state.

Further, assumption of the goodness of a given density functional can be estimated by comparing the results with the experimental values that are being reproduced or with the results that can be obtained through post Hartree–Fock calculations such as MP2, MP4 or CCSD. However, this is not always possible due to the lack of experimental results for the molecular systems that are being studied or the large size of the molecules that keep some accurate methodologies computationally practical. For this reason, we have developed a protocol named Koopmans in DFT (KID) [92,93,94], which is an attempt to validate a given density functional in terms of its internal coherence. Within the KID protocol, four descriptors have been defined where it has been shown that there is a connection between those descriptors and the simplest conformity to the theorem of Koopmans or the Ionization Energy theorem, which is its equivalent within the Generalized Kohn–Sham (GKS) version of DFT, by connecting the electronic energy of the Highest Occupied Molecular Orbital (HOMO)—that is, $\varepsilon_{H}$—with the negative if the ionization energy, I, and the electronic energy of the Lowest Unoccupied Molecular Orbital (LUMO)—that is $\varepsilon_{L}$—with the negative of the Electron affinity, A, giving rise to $J_{I}=\left|\varepsilon_{H}+E_{gs}\left(N-1\right)-E_{gs}\left(N\right)\right|$, $J_{A}=\left|\varepsilon_{L}+E_{gs}\left(N\right)-E_{gs}\left(N-1\right)\right|$, and $J_{HL}=\sqrt{J_{I}^{2}+J_{A}^{2}}$, where $E_{gs}\left(N\right)$, $E_{gs}\left(N-1\right)$ and $E_{gs}\left(N+1\right)$ represents the ground state energy of the neutral molecule, of the radical cation and the radical anion, respectively, in the geometry of the neutral molecule. An additional descriptor, ∆SL, was designed [92,93,94] to help in the verification of the accuracy of the KID approximation by comparing the HOMO energy of the radical anion with the energy of the LUMO of the neutral species. Although the Koopmans complaint behavior of the MN12SX density functional has been proven previously for the case of peptides [92,93,94], we think that it is worth performing a further validation for the case of the molecules considered in the present study.

As we have shown in our previous research [85,86,87,88,89,90,95], the KID procedure is also valid in the presence of water as the solvent and represents an advantage over the use of the vertical I and A for the calculation of the global descriptors because it avoids the separate calculation of the radical cation and anion which could be difficult for molecules of the size considered here. It has been shown by Frau and Glossman-Mitnik [69,84,85,86,87,88,89,90,95] that the HOMO and LUMO energies obtained with the MN12SX/Def2TZVP/H2O model chemistry allow the verification of the KID procedure—that is, rendering an approximate Koopmans behavior. All the calculated molecules are neutral—that is, they have a charge equal to zero—and these optimized structures were considered for the estimation of the energies of the cation and anion radicals because the definitions of the conceptual DFT descriptors have been derived at constant external potential [72,73,74,96,97].

The HOMO, LUMO and Singly Occupied Molecular Orbital (SOMO, which is equivalent to the HOMO of the radical anion) orbital energies, HOMO–LUMO gap and the KID descriptors (all in eV) tested in the verification of the Koopmans-like behavior of the MN12SX density functional for the new BQ molecules are shown in Table 3, while the calculated values for these global reactivity descriptors using the MN12SX/Def2TZVP/H2O model chemistry and the KID procedure are displayed in Table 4.

It can be seen from the results in Table 1 that the descriptors considered for the estimation of the goodness of the selected density functional through the KID procedure are very close to zero for all the studied molecules, providing an accurate justification for the choice of the MN12SX/Def2DZVP/H2O model chemistry employed for the computational determinations in this study.

As a complement to these global reactivity descriptors that arise from conceptual DFT, Domingo and his collaborators [98] proposed a nucleophilicity index (N) (not to be confused with the number of electrons) through the consideration of the HOMO energy obtained through the KS scheme with an arbitrary shift of the origin taking the molecule of tetracyanoethylene (TCE) as a reference. On the basis of the previous definition and the scale established by these authors [98], and the results presented in Table 4, it can be concluded that all the molecules considered in this study can be regarded as moderate nucleophiles.

Next, QSAR analysis was carried out on the molecular properties of the new BQ molecules, such as ∆G of solvation, molecular weight, Log P, total polar surface area (TPSA), and molecular volume, using Molinspiration software. The predicted parameters are shown in Table 5.

The results presented in Table 5 can be better understood if a graphical representation of these physicochemical properties displays in the form of what is known as a bioavailability radar, where six physicochemical properties are considered: lipophilicity, size, polarity, solubility, flexibility, and saturation. Each descriptor must fall entirely within the physicochemical range depicted in the pink area of the radar plot of the molecule to be considered drug-like. These results are presented in Figure 7.

Next, a web tool known as SwissTargetPrediction for efficient prediction of protein targets of small molecules was considered for the determination of the potential bioactivity of the molecules considered in this study [99]. The associated website allows the estimation of the most probable macromolecular targets of a small molecule assumed as bioactive. The prediction confirms that a combination of 2D and 3D similarity with a library of 370,000 known actives on more than 3000 proteins from three different species, as shown in Figure 8.

During the development process of a new drug, it is very important to know the possible fate of a therapeutic compound in an organism, a process that is known as pharmacokinetics. This can be achieved by analyzing the associated effects in the form of individual indices that involve Absorption, Distribution, Metabolism, and Excretion (ADME) parameters. These parameters can be obtained by using computer models that can be alternatives to the experimental procedures for their determination. In this work, some ADME parameters were estimated with the aid of the SwissADME software that is available online [100], and the results are presented in Table 6.

## 3. Materials and Methods

### 3.1. Analysis of Minimum Inhibitory Concentration against C. albicans

The minimum inhibitory concentrations (MICs) were calculated according to the proposed protocol by Schwalve et al. (2007). Briefly, 20 µL of each pyrrolo[1,2-a]quinoline derivative from the stock solution was added into the first tube containing 380 µL of brain heart infusion (BHI, HiMedia) broth to achieve 100 µg/mL concentration. Then, 200 µL of each derivative was serially diluted starting from 100 to 0.2 µg/mL (in total of 9 dilutions i.e., 10^−1^ to 10^−9^) in BHI broth separately. From the stock solution, 5 µL of inoculum containing pure culture of *C. albicans* (ATCC no. 10231; 16 µg/mL) was added into 2ml of BHI broth and later, mixed with the serially diluted tubes containing pyrrolo[1,2-a]quinoline derivatives. The standard value for fluconazole in the MIC tests was 30 µg. The tubes were later incubated for 24 h and observed for appearance of turbidity.

### 3.2. Selection of Protein

The synthesized ligand compounds act as inhibitors that reduces the growth of *C. albicans* by interacting with the pathogenic proteins such as secreted aspartic protease 3 (SAP3), fungal cell surface protein (β-glucanase) and the sterol 14-alpha demethylase receptors as reported elsewhere. In this study, the protein targets identified are SAP protein (SAP-3), surface protein and sterol 14-alpha demethylase, which were used to carry out the in silico analyses. The required crystal structures of the above-mentioned proteins of *C. albicans* were selected and extracted from an online Protein Data Bank (PDB) database based on their structures, functions and resolutions. The PDB IDs for the SAP-3 protein, β-glucan protein and sterol 14-alpha demethylase are 2H6T, 3N9K and 5TZ1, respectively, and a further file was secured in pdb format for further analysis. The molecular visualization of these proteins was accomplished by utilizing Chimera software [101] and required modifications were applied to remove the nonstandard amino acids or the pre-existing ligands and the water molecules attached with the protein structure (Figure 9a–c). Next, the stability of the protein structures was assessed using RAMPAGE [102], which displays the percentage of amino acid residues present in the allowed and favored regions. The proteins with more than 96% of residues in the favored and less than 2% of residues in the allowed regions were selected for further in silico analysis. Based on the desirable value obtained, the selected pathogenic proteins for further molecular interaction and docking analysis were macromolecules.

### 3.3. Small Molecule Optimization

The structures of small molecules (BQ-01, BQ-03, BQ-05 and BQ-07) used as molecular targets against *C. albicans* were saved in the pdb format for further in silico analysis. For preparation of ligand, we utilized three different tools: 1. Chemsketch software [103], used here to sketch the small molecules and the resulted output file was saved in chemical markup language (cml) format; 2. OpenBabel software [104], utilized for conversion of the 2D structure cml file obtained from Chemsketch to pdb format, thereby generating the 3D coordinates; 3. Chimera software, utilized for visualization of resultant file obtained from OpenBabel (Figure 10a–d). The small molecules referred to as ligands in the above mentioned file format were subjected to further in silico molecular interaction analysis.

Next, comparative analyses were carried out to compare the binding interactions results from the synthesized ligands and the pre-existing drug against the *C. albicans* surface and SAP-3 receptors. The .sdf format file of the FDA approved drug file was taken from the online database PubChem. The mandatory .pdb 3D format file for the in silico docking analyses is essential; hence, the .sdf format of the drug was converted to .pdb format using OpenBabel software and visualized using Chimera software (Figure 11).

### 3.4. In Silico Molecular Interaction and Docking

The molecular interaction of the small molecules and their inhibitory potential against the *C. albicans* fungal proteins were studied using in silico molecular docking. Here, the molecular docking interactions were between the selected proteins from PDB against the optimized small molecules using PyRx software [105]. The respective macromolecule and the ligands were loaded onto the software and the active site residues were selected and grid box was generated around the selected amino acid residues. The docking results was carried out using a genetic algorithm. Moreover, the interaction achieved eight docking confirmations.

### 3.5. Conceptual DFT Reactivity Descriptors of the Molecules and Their Related Pharmacokinetics

The determination of the conformers of the molecules considered in the current study was performed by resorting to MarvinView 17.15, which is available from ChemAxon (http://www.chemaxon.com, accessed on 6 October 2020), by carrying out molecular mechanics calculations through the overall MMFF94 force field [75,76,77,78,79]. This was followed by a new geometry optimization and frequency calculation by means of the Density Functional Tight Binding (DFTBA) methodology [80]. This last step was required to verify the absence of imaginary frequencies to assess for the minimum stability of the optimized structures-in the energy landscape. The electronic properties and the chemical reactivity descriptors of the studied molecules involved the use of MN12SX/Def2TZVP/H2O [81,82,83] model chemistry on the previously optimized molecular structures because it has been shown that it allows the verification of the “Koopmans in DFT” (KID) procedure [69,84,85,86,87,88,89,90] with the aid of the Gaussian09 software [80] and the SMD solvation model [91].

With the aid of the KID technique and the finite difference approximation [69,84,85,86,87,88,89,90,95], the following expressions can be used to define the global reactivity descriptors [72,73,74,96,97]:

Electronegativity
(1)χ= −12I+A≈12εL+εL

Chemical Hardness
(2)$$\eta =\;\left(I-A\right)\approx \left(\varepsilon_{L}-\varepsilon_{H}\right)$$

Global Electrophilicity
(3)$$\omega =\;\frac{\chi^{2}}{2\eta }=\;\frac{{\left(I+A\right)}^{2}}{4\left(I-A\right)}\approx \frac{{\left(\varepsilon_{L}+\varepsilon_{H}\right)}^{2}}{4\left(\varepsilon_{L}-\varepsilon_{H}\right)}$$

Electrodonating Power
(4)$$\omega^{-}=\;\frac{{\left(3I+A\right)}^{2}}{16\left(I-A\right)}\approx \frac{{\left(3\varepsilon_{H}+\varepsilon_{L}\right)}^{2}}{16\eta }$$

Electroaccepting Power
(5)$$\omega^{+}=\;\frac{{\left(I+3A\right)}^{2}}{16\left(I-A\right)}\approx \frac{{\left(\varepsilon_{H}+3\varepsilon_{L}\right)}^{2}}{16\eta }$$

Net Electrophilicity
(6)$$∆\omega^{\pm }=\;\omega^{+}-\left(-\omega^{-}\right)=\;\omega^{+}-\omega^{-}$$
where, as mentioned, $\varepsilon_{H}$ and $\varepsilon_{L}$ are the energies of the HOMO and LUMO, respectively.

After analyzing KID descriptors, The SMILES notation of all the studied compounds were fed into the online Molinspiration software from Molinspiration Cheminformatics (http://www.molinspiration.com, accessed on 6 October 2020) for the calculation of the molecular properties (∆G of Solvation, molecular weight, Log P, total polar surface area (TPSA), and molecular volume). The bioactivity scores were compared to those obtained using other software such as MolSoft from Molsoft L.L.C. (http://molsoft.com/mprop/, accessed on 6 October 2020) and ChemDoodle Version 9.02 from iChemLabs L.L.C. (http://www.chemdoodle.com), accessed on 6 October 2020.

## 4. Conclusions

The study shows novel pyrrolo[1,2-a]quinoline derivatives as putative drug targets of *C. albicans*. Here, in vitro screening of novel analogue derivatives against *C. albicans* fungal pathogen displayed inhibitory potentials of 75-fold for BQ-06, 07, and 08, and 37.5-fold for BQ-01, 03, and 05, followed by 18.75-fold for BQ-04, and 2.4-fold for BQ-02, in comparison to the standard antifungal fluconazole. Further, in silico molecular docking prediction of the ligand molecules BQ-01, 03, 05 and 07 compared to the drug alone exhibited strong hydrogen binding interactions with the regulating C. *albicans* protein β-glucan (3N9K), which demonstrates thermodynamically favorable conditions. Furthermore, these compound shows high binding energies ranging from −6.5 to −9.7 kcal/mol. However, based on the chemical structure, BQ-01, 03, 05 and 07 consist of methyl ester groups at the second and third positions compared to that of only one methyl ester at the third position, which is the case for BQ-02, 04, 06 and 08. Furthermore, BQ-01, 03, 05 and 07 possess a two-electron donor group, due to their effective interaction with C. *albicans* fungal proteins, which inhibits their growth compared to the single-electron donor groups present at the third positions of BQ-02, 04, 06 and 08. In addition, the determination of the chemical reactivity descriptors that arise from conceptual DFT allowed us to quantify the different values for the chemical reactivity of the studied molecules. Further, QSAR parameters, biological radars, exploration of possible targets and the pharmacokinetics of the systems suggested that compounds BQ-01, 03, 05 and 07 have good drug-like properties. In conclusion, the analyzed parameters can be useful in complementing the experimental data as the starting point for the development of new antifungal therapeutic drugs.

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
