# Peer review of "Investigation of Antifungal Properties of Synthetic Dimethyl-4-Bromo-1-(Substituted Benzoyl) Pyrrolo[1,2-a] Quinoline-2,3-Dicarboxylates Analogues: Molecular Docking Studies and Conceptual DFT-Based Chemical Reactivity Descriptors and Pharmacokinetics Evaluation"

_molecules, 2021, doi:10.3390/molecules26092722_

Round 1
Reviewer 1 Report
This manuscript describes a series of eight pyrrolo [1, 2-a]quinoline derivatives for potential antifungal properties. The described study is a continuation of the authors’ already published work Ref. [44] V. Uppar, K. K. Mudnakudu-Nagaraju, A. I. Basarikatti, M. Chougala, S. Chandrashekharappa, M. K. Mohan, G. Banuprakash, K. N. Venugopala, R. Ningegowda, B. Padmashali. Microwave Induced Synthesis, and Pharmacological Properties of Novel 1-benzoyl-4-bromopyrrolo[1,2-a]quinoline- 3-carboxylate Analogues. Chemical Data Collections 25 (2020) 100316. doi:10.1016/j.cdc.2019.100316.
The same eight compounds were tested for their antibacterial activity against gram-positive and gram-negative bacterial strains, what the authors should have clearly noted in the present report.
Nevertheless, this manuscript contains interesting and probably useful information. The characteristics of the new research (Molecular Docking, Determination of the Conceptual DFT Reactivity Descriptors of the Molecules and their Related Pharmacokinetics) should be more detailed and most of all explained, not just mentioned.
For example, The interpretation of results of molecular docking should be presented in Figure 5-9 in detailed, The compound interacts with residues with the formation of hydrogen bonds – indicate the location and exact elements of the bond. It is important to note that the compounds of the presented series differ from each other or not. This aspect of the present study is important and both large and small differences must be explained.
The manuscript is generally well written and organized, but authors should consider before resubmitting the information they wish to convey thanks to drawings and Tables:
(i) Figure 1- select biologically active moiety of the molecule
(ii) Figure 2 - The PDB structure of the three selected proteins gives the reader little information because of small differences in drawings and Figure 3 - The 3-dimensional structure of the ligand molecules visualized - differences that are difficult to distinguish
(iii) improve the quality of Figures 11, the legend should be more visible
(iv) Figure 5-9 - the same content repeated too many times
(v) Table 6 - define the content: “Yes”, “No” and “High”- additionally written in capital letters
The choice of the journal will be correct and such work may be published in Molecules, but authors before resubmission should consider some changes in the manuscript in accordance with the comments.
Author Response
Response to Reviewer #1
Comment 1: This manuscript describes a series of eight pyrrolo [1,2-a]quinoline derivatives for potential antifungal properties. The described study is a continuation of the authors’ already published work Ref. [44] V. Uppar, K. K. Mudnakudu-Nagaraju, A. I. Basarikatti, M. Chougala, S. Chandrashekharappa, M. K. Mohan, G. Banuprakash, K. N. Venugopala, R. Ningegowda, B. Padmashali. Microwave Induced Synthesis, and Pharmacological Properties of Novel 1-benzoyl-4-bromopyrrolo[1,2-a]quinoline- 3-carboxylate Analogues. Chemical Data Collections 25 (2020) 100316. doi:10.1016/j.cdc.2019.100316.
The same eight compounds were tested for their antibacterial activity against gram-positive and gram-negative bacterial strains, what the authors should have clearly noted in the present report.
Response:
Thank you for valuable suggestion. We have stated and highlighted the antibacterial activity of the eight compounds against gram-positive and gram-negative bacterial strains in the introductory part of the revised version of the manuscript.
Comment 2: Nevertheless, this manuscript contains interesting and probably useful information. The characteristics of the new research (Molecular Docking, Determination of the Conceptual DFT Reactivity Descriptors of the Molecules and their Related Pharmacokinetics) should be more detailed and most of all explained, not just mentioned.
Response:
We have included additional information to the revised and corrected version of the manuscript with a more detailed explanation of the KID descriptors and the significance of their values for the case of the molecules under study. Moreover, in the conclusions section, there is a mention to the potential usefulness of the information of the Conceptual DFT reactivity descriptors as well as the pharmacokinetics for the potential design of new drugs based on the studied molecules.
Comment 3: For example, the interpretation of results of molecular docking should be presented in Figure 5-9 in detailed, the compound interacts with residues with the formation of hydrogen bonds – indicate the location and exact elements of the bond. It is important to note that the compounds of the presented series differ from each other or not. This aspect of the present study is important and both large and small differences must be explained.
Response:
As per the reviewer suggestion, the interpretation of results of molecular docking for the 8 pyrrolo [1, 2-a]quinoline derivatives (presented in Fig. 2-6) are incorporated in the result and discusión section of the maunscript. The possible explantion stated as “out of 8 pyrrolo [1, 2-a]quinoline derivatives, 4 have chosen for molecular docking study, which are BQ-01, BQ-03, BQ-05, and BQ-07. These molecules sharing a common methyl ester group at 2nd and 3rd position instead of the hydrogen group at 2nd and ethyl ester at 3rd position respectively of BQ-02, BQ-04, BQ-06 and BQ-08. Further, the hypotheses suggests that the two-electron donor group present in BQ-01, 03, 05 & 07 might possibly have a stronger interaction with C. albicans proteins and inhibits its growth compared to single electron donor group on the BQ-02, 04, 06 and 8”.
Comment 4: The manuscript is generally well written and organized, but authors should consider before resubmitting the information they wish to convey thanks to drawings and Tables:
(i) Figure 1- select biologically active moiety of the molecule
(ii) Figure 2 - The PDB structure of the three selected proteins gives the reader little information because of small differences in drawings and Figure 3 - The 3-dimensional structure of the ligand molecules visualized - differences that are difficult to distinguish
(iii) improve the quality of Figures 11, the legend should be more visible
(iv) Figure 5-9 - the same content repeated too many times
(v) Table 6 - define the content: “Yes”, “No” and “High”- additionally written in capital letters.
Response:
Thank you for the comment. In the revised version of the manuscript, we have included appropriate references and acknowledged wherever applicable.
(i) Highlighted the biologically active moiety of the molecule in Figure 1 and added an explanation for the highlighted parts in legend of the Figure 1. These corrections have been incorporated into the revised version of the manuscript. The explanation given in the legend of Figure 1 is as follows: “All the eight pyrrolo[1,2-a]quinoline derivatives have a common bromo-substituent at 4th position. The four compounds namely BQ-01, BQ-03, BQ-05 and BQ-07 consists of a methyl ester group at 2nd and 3rd position compared to hydrogen group at 2nd and ethyl ester at 3rd position respectively of BQ-02, BQ-04, BQ-06 and BQ-08. Further, BQ-01, 03, 05 and 07 have two ester groups on 2nd and 3rd position (as highlighted), which donates two electrons whereas only one electron donor group is present in the BQ-02, 04, 06 and 08 derivatives”.
(ii) The Figure 2 and 3 in the submitted manuscript has been changed as Figure 9 and 10 respectively in the revised version of manuscript. The 3-dimensional structure of the ligands in Figure 10 has been changed into more distinguishable way.
(iii) Figure 11 in the submitted manuscript has been changed into Figure 8 in the revised version of manuscript.
(iv) Figure 5-9 in the submitted manuscript has been changed as into Figure 2-6 In the revised version of manuscript (Figure 5, 6, 7, 8 and 9 as Figure 2, 3, 4, 5, and 6 respectively) and the repeated contents has been removed and corrected figure legend.
(v) The content of the Table 6 has been defined and corrected to capital letters in the revised version of manuscript.
Comment 5: The choice of the journal will be correct and such work may be published in Molecules, but authors before resubmission should consider some changes in the manuscript in accordance with the comments.
Response: We thank the Reviewer for his positive comments that helped to improve our manuscript significantly.

Reviewer 2 Report
The manuscript brings an interesting information on the title compounds. It is worth of publishing but some improvements are necessary. My comments may be summarized as follows:
- All acronyms must be explained at their first occurence. Some terms are never explained (greek mi, log P, MW, TPSA, SOMO, MN12SX/DefTZVP/H2O, Egs, E(N) ).
- Various symbols are used for the same quantities (EL and L, EH and H ...).
- N is used for nucleofility (not defined) as well as for the number of electrons in E(N) etc.
- The charges (neutral?) and spin states (singlet?) of the systems under study are not mentioned.
- It is not mentioned, that unlike HOMO and LUMO of neutral systems in the ground singlet spin state, SOMO is related to the anionradical in the doublet spin state (alpha-HOMO)
- How did you check the stability of the optimized geometries?
- What geometry has been used for anionradicals?
- The DFT functional and basis sets are not referenced.
- All softwares used must be referenced.
- The equations at lines 261 – 276 are not numbered. Some signs are misssing. The electronegativity definition is incorrect.
- The discussion of the calculated reaction and KID descriptors is desirable.
- Some misprints and grammatical errors can be found by more careful reading. Capitals writing must be revised (Kcal/Mol etc.).
Finally it can be concluded that this manuscript demands major revision to be published.
Author Response
Response to Reviewer #2
The manuscript brings an interesting information on the title compounds. It is worth of publishing, but some improvements are necessary. My comments may be summarized as follows:
- All acronyms must be explained at their first occurrence. Some terms are never explained (Greek mi, log P, MW, TPSA, SOMO, MN12SX/DefTZVP/H2O, Egs, E(N) ).
Response:
All the acronyms has been explained at their first occurrence in the revised version of the manuscript.
- Various symbols are used for the same quantities (EL and L, EH and H ...).
Response:
This problem has been solved and corrected in the revised version of the manuscript.
- N is used for nucleofility (not defined) as well as for the number of electrons in E(N) etc.
Response:
The definition for the Nucleophilicity N, the corresponding reference and the analysis of their values have now been included in the revised and corrected version of the manuscript. A note has been included explaining that it must not be confused with the number of electrons.
- The charges (neutral?) and spin states (singlet?) of the systems under study are not mentioned.
Response:
This information is now presented in the revised and corrected version of the manuscript.
- It is not mentioned that unlike HOMO and LUMO of neutral systems in the ground singlet spin state, SOMO is related to the anion radical in the doublet spin state (alpha-HOMO)
Response:
This information is now presented in the revised and corrected version of the manuscript.
- How did you check the stability of the optimized geometries?
Response:
An explanation about the verification of the stability of the optimized geometries has now been included in the revised and corrected version of the manuscript.
- What geometry has been used for anion radicals?
Response:
For the determination of the electronic energies of the cation and anion radicals, it has been considered the geometry of the neutral ground state, because the definitions of the Conceptual DFT descriptors implies a variation at constant external potential. This explanation has now been included in the revised and corrected version of the manuscript.
- The DFT functional and basis sets are not referenced.
Response:
The DFT density functional and basis set are now referenced in the revised and corrected version of the manuscript.
- All the software used must be referenced.
Response:
All the software used is now referenced in the revised and corrected version of the manuscript.
- The equations at lines 261 – 276 are not numbered. Some signs are missing. The electronegativity definition is incorrect.
Response:
There was a problem with the typing of the equations. They have been revised and corrected, and a sequential number has been added.
- The discussion of the calculated reaction and KID descriptors is desirable.
Response:
A better information about the KID descriptors and their results has now been included in the revised and corrected version of the manuscript.
- Some misprints and grammatical errors can be found by more careful reading. Capitals writing must be revised (Kcal/Mol etc.).
Response:
We have done our best in correcting all misprints and grammatical errors that were present in the previous version and we hope that the revised and corrected version of the manuscript is free of any mistake.
Finally, it can be concluded that this manuscript demands major revision to be published.
Response:
We thank the Reviewer for his comments that helped to improve our manuscript.

Round 2
Reviewer 2 Report
i) The details of quantum-chemical calculations are presented in Results and Discussion (section 2.3) instead of Materials and Methods.
ii) insert 'in the geometry of the neutral molecule' at the end of the sentence in line 415.
iii) Table 6: Despite Kp is in cm/s, log Kp is dimensionless. I propose to use log(Kp/cm s-1) in the 1st column.
iv) Several misprints and grammatical errors need correction (mostly plural vs singular etc.)
Author Response
Response to Reviewer Round #2
We thank the reviewer for providing comments and suggestions, which has improved the current version of the manuscript. We provide the following responses to the reviewer comments as follows:
Comment: i) The details of quantum-chemical calculations are presented in Results and Discussion (section 2.3) instead of Materials and Methods.
Response: As per the suggestions, the details of the quantum-chemical calculations have now been described in the Material and Methods section 3.5. We as well moved the required information to Materials and Methods section 3.5 of the revised manuscript from section 2.3.
Comment: ii) insert 'in the geometry of the neutral molecule' at the end of the sentence in line 415.
Response: The suggestion has been incorporated in the revised version of the manuscript.
Comment: iii) Table 6: Despite Kp is in cm/s, log Kp is dimensionless. I propose to use log(Kp/cm s-1) in the 1st column.
Response: The suggestion has been incorporated in the revised version of the manuscript.
- iv) Several misprints and grammatical errors need correction (mostly plural vs singular etc.)
Response: We have corrected and included the misprints and as well checked for grammatical errors in the revised version of the manuscript. The corrections are highlighted in yellow.
